# Use of a DNN-Based Image Translator with Edge Enhancement Technique to Estimate Correspondence between SAR and Optical Images

**Hisatoshi Toriya [1,2,*]**, **Ashraf Dewan [3]**, **Hajime Ikeda [1]**, **Narihiro Owada [1]**, **Mahdi Saadat [1]**, **Fumiaki Inagaki [1]**, **Youhei Kawamura [4]** and **Itaru Kitahara [2]**

1   Faculty of International Resource Sciences, Akita University, 1-1 Tegatagakuen-Machi, Akita-City 0100862, Akita, Japan; ikeda@gipc.akita-u.ac.jp (H.I.); owada@gipc.akita-u.ac.jp (N.O.); mahdi.saadat1@gipc.akita-u.ac.jp (M.S.); fumiaki.inagaki@gipc.akita-u.ac.jp (F.I.)
2   Center for Computational Sciences, University of Tsukuba, 1-1-1 Tennodai, Tsukuba-City 3058577, Ibaraki, Japan; kitahara@ccs.tsukuba.ac.jp
3   School of Earth and Planetary Sciences, Curtin University, Kent St. Bentley, WA 6102, Australia; a.dewan@curtin.edu.au
4   Faculty of Engineering, Hokkaido University, Kita 8, Nishi 5, Kita-Ku, Sapporo-City 0608628, Hokkaido, Japan; kawamura@eng.hokudai.ac.jp
*   Correspondence: toriya@gipc.akita-u.ac.jp

**Abstract:** In this paper, the local correspondence between synthetic aperture radar (SAR) images and optical images is proposed using an image feature-based keypoint-matching algorithm. To achieve accurate matching, common image features were obtained at the corresponding locations. Since the appearance of SAR and optical images is different, it was difficult to find similar features to account for geometric corrections. In this work, an image translator, which was built with a DNN (deep neural network) and trained by conditional generative adversarial networks (cGANs) with edge enhancement, was employed to find the corresponding locations between SAR and optical images. When using conventional cGANs, many blurs appear in the translated images and they degrade keypoint-matching accuracy. Therefore, a novel method applying an edge enhancement filter in the cGANs structure was proposed to find the corresponding points between SAR and optical images to accurately register images from different sensors. The results suggested that the proposed method could accurately estimate the corresponding points between SAR and optical images.

**Keywords:** image registration; keypoint matching; synthetic aperture radar; deep neural network; generative adversarial networks

## 1. Introduction

When a natural disaster such as an earthquake or tsunami occurs, visual information can provide essential data for emergency management. Aerial images obtained from satellites, aircraft, and drones can simultaneously capture a wide range of features; thus, they may be utilized for response and recovery operations after a disaster event [1,2]. The higher the shooting altitude, the wider the view; however, the visibility of optical images can deteriorate owing to the lack of a light source or the influence of clouds. Alternatively, synthetic aperture radars (SARs) can capture data over a large area without much deterioration. Therefore, SARs are often used in disaster situations, such as damage area detection [3–5], infrastructure damage assessment [6,7], etc. However, SAR has some problems in practical use, including (a) they are relatively less readable by humans and (b) landmark points may be difficult to locate, especially with coarse resolution data. To obtain geographic information, the combined use of SAR images and optical data can provide a wealth of information. Thus, the geometric registration of the two data is essential, which may be challenging. The accurate registration of SAR images and optical data can

provide valuable information, which would otherwise be difficult, especially during natural disasters, such as floods of large magnitude. Hence, the precise registration of optical and microwave data can support rapid scanning of flood areas or the identification of collapsed buildings after an earthquake or tsunami for rescue and evacuation efforts.

The pixel values in SAR images represent the intensity of electromagnetic waves, and they are expressed in backscatter values. This differs from visible light, thereby resulting in varying responses of several ground features. Thus, when handling SAR images, it is common to compensate for their low readability by registering them with optical images.

Image registration is traditionally performed using a digital elevation model (DEM) [8]. However, when the spatial resolution of both the optical and SAR images is high, DEM may not be effective. Moreover, when a disaster event occurs, landforms may be drastically changed such that DEM may not achieve accurate image registration. An alternative method is image-based registration, which is independent of DEM. In this method, complementing factors associated with the acquisition and processing of optical and SAR data are considered because optical images are prone to atmospheric disturbances, especially during disasters such as floods, while SAR can acquire data in any weather. Combining the SAR data acquired during the disaster and the optical images before/after the disaster can make response and recovery operations much easier. Meanwhile, an automated method for the geometric rectification of images from different sensors is not as easy due to variations in the spectral response of ground features. Hence, two datasets can be combined by translating the appearance of a SAR image to an optical image using generative adversarial networks (GANs) [9] for subsequent keypoint matching.

We proposed a method for finding local feature correspondence between multimodal (SAR and optical) images using an image-based feature keypoint extraction, description, and matching algorithm [10], as shown in Figure 1. Image translation with a GAN is used to transform a SAR image into an optical image. Although this method has good accuracy, the blurring of features could be a significant issue (Figure 2). To obtain more corresponding points, blurring should be removed to highlight the local features. However, it is difficult to achieve this with conventional GANs.

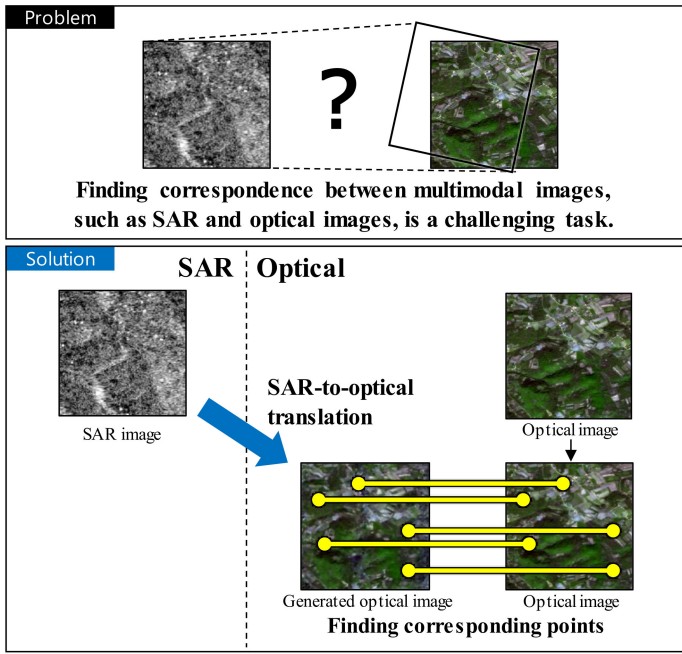

**Figure 1.** Outline of the proposed method. Using a GAN as a pre-processing step before keypoint matching, local correspondence was established for multimodal image registration.

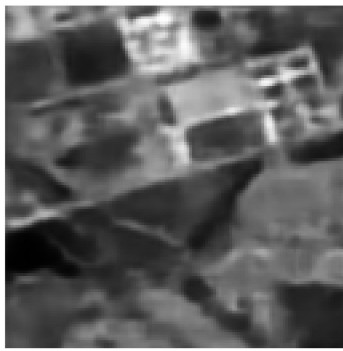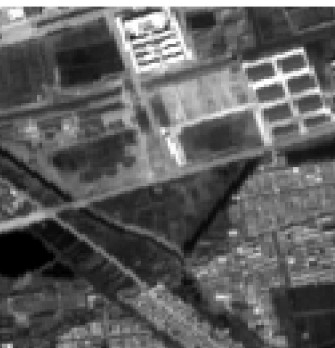

**Figure 2.** An optical image generated by a conditional GAN ([11], **left**) and an optical image (**right**). Note the loss of local features due to blurring.

In this work we proposed a SAR and optical image registration method for overcoming the above-mentioned problem. By training the GAN with an edge-enhancement technique, we could obtain an image translator (generator) with higher quality than a conventional GAN. Furthermore, comparative evaluation was conducted to quantitatively analyze the effectiveness and performance of the proposed method. Meanwhile, local features were necessary for finding the corresponding points, and they could be obtained by applying an edge-enhancement filter. The main contributions of this work are as follows.

a   A novel method was proposed for finding the corresponding points between SAR and optical images for image registration.

b   The SAR-to-optical image translator was improved by training a GAN with edge-enhanced images to maintain local image features and improve keypoint extraction.

c   The efficiency of the proposed method was demonstrated by a comparison experiment with conventional methods and a qualitative experiment.

This paper is organized as follows. Conventional image-registration methods for multimodarl images are discussed in Section 2, as well as image translation methods with GANs. In Section 3, we describe how to train a GAN to obtain a generator that can perform SAR-to-optical-image translation. The methods for finding the corresponding points between the optical and generated optical images are described in Section 4. The experimental setting, results, and discussions are given in Section 5. Finally, the conclusion and major findings of this work are discussed in Section 6.

## 2. Related Work

An image-based (not DEM-based) registration method is expected to match multi-modal images based on the similarities of their edges and corners. Thus, template-based methods with traditional metrics, such as normalized cross-correlation (NCC) or mutual information between two images, have been proposed for image registration [12,13]. Meanwhile, other suitable metrics have been proposed for more accurate template-matching of SAR and optical images [14–17]. However, these methods use image features of a relatively wide area. The template-matching accuracy decreases with small template size, whereas its robustness for occlusions and partial difference (e.g., between pre- and post-disaster) decreases with large template size. Although machine-learning-based (especially deep-learning-based) SAR and optical image-matching methods have been proposed [18–20], the range of pixels considered in these methods are whole image correspondences or some of them have a limitation in rotation robustness. The identification of local correspondence is not possible with these methods; hence, a new technique that can perform image registration with local features when applied to disaster sites is required since landform responses may be partially or fully changed after a disaster event.

Local image features are often employed to achieve the image-based registration of satellite data. Particularly, keypoint-based methods [21–23] can estimate the correspondence between two images using uncorrelated local features, similar to template-based

methods. Methods that adopt a keypoint detector and a feature descriptor [24–27] are frequently used to match (i.e., estimate the correspondence of) two images. These methods describe features that are robust to geometric fluctuation (e.g., rotation, scaling) and changes in environmental (e.g., lighting) conditions. They find keypoint pairs that have similar features as corresponding points, thereby achieving image matching. Since these methods use local features, they can achieve partial correspondence, even when a part of the captured area has collapsed due to disaster.

Meanwhile, machine learning methods based on deep neural networks (DNNs) have been widely used for image modal translation [28–32]. In this work, we mainly focus on GANs. A GAN trains a generator that generates data and a discriminator that determines the authenticity of the data to produce a generator that can generate data similar to the original features through comparison. The loss function of a GAN, $\mathcal{L}_{\text{GAN}}$, is given as follows:

$$\mathcal{L}_{\text{GAN}}(G, D) = \mathbb{E}_{\boldsymbol{y}}[logD(\boldsymbol{y})] + \mathbb{E}_{\boldsymbol{z}}[log(1 - D(G(\boldsymbol{z})))], \tag{1}$$

where $G()$ indicates the generated data based on the input data by the generator ($G$), $D()$ indicates the probability that a discriminator ($D$) can correctly discriminate between a real input and a fake (artificial) input, $\boldsymbol{y}$ and $\boldsymbol{z}$ indicate the answer and a random value, respectively. The purpose of training a GAN is to obtain a well-trained generator that can produce sufficient fake data to deceive the discriminator. The generator $G^*$ is given as follows:

$$G^* = arg \min_{G} \max_{D} \mathcal{L}_{\text{GAN}}(G, D). \tag{2}$$

An application of a GAN is a conditional GAN (cGAN), whose generator obtains inputs rather than random values. It is a common multipurpose image interpretation method, and its loss function $\mathcal{L}_{\text{cGAN}}$ is given as follows:

$$\mathcal{L}_{\text{cGAN}}(G, D) = \mathbb{E}_{\boldsymbol{y}}[logD(\boldsymbol{y})] + \mathbb{E}_{\boldsymbol{x,z}}[log(1 - D(\boldsymbol{x}, \ G(\boldsymbol{x}, \boldsymbol{z})))], \tag{3}$$

where $\boldsymbol{x}$ indicates the input. A well-trained generator for the cGAN can be obtained in the same way as Equation (2). It has been demonstrated that cGAN permits multimodal image-to-image translation, e.g., from an artificial room image to a real photo [33], or a sketched image to a real photo [11].

Another advantage of a GAN is that fewer training datasets are required [11]. Although a large number of training datasets is necessary for conventional machine learning methods, GANs can achieve high performance with fewer datasets owing to their generator and discriminator models. This was a crucial advantage for this study because it was difficult to prepare several datasets of aligned SAR and optical images under a disaster situation.

Therefore, we proposed a method for performing SAR-to-optical image translation using a GAN so that a keypoint-matching algorithm could be applied to multimodal images. By applying cGAN to SAR-to-optical image translation, the challenging task of multimodal image registration was reduced to the conventional task of monomodal image registration, which could be solved using a feature-based image-matching algorithm.

Blurring, as shown in Figure 2, is one of the factors that reduces the accuracy of matching. Edge enhancement for super-resolution [34] and that for small object detection [35] have been proposed. Although these methods include edge enhancement in the neural network structures, they increase the complexity of implementation, such as parameter tuning from applying the methods to problems. Therefore, in this paper, we used a neural network structure, which was used in previous studies and its performance had been established. The edge enhancement was used to pre-process the image, which simplified the implementation and improved the performance.

### 3. Training a GAN for SAR-to-Optical Image Translation with Edge Enhancement

In this section, the process of training a GAN to generate optical images (generated optical images) from SAR images is described. The generated optical images were obtained by machine-learning-based prediction using SAR images as input. Figure 3 shows the cGAN training model, in which the generator $G$ and discriminator $D$ are combined to obtain higher-quality generated optical images. In preparing SAR and optical image pairs that were already co-registered, we set the SAR images as input $x$ to the model and $y$ as the correct answer for training $G$ and $D$. With this SAR-to-optical image translation process, the generated and original optical images had the same modality, and it was possible to perform image-registration processing with keypoint matching.

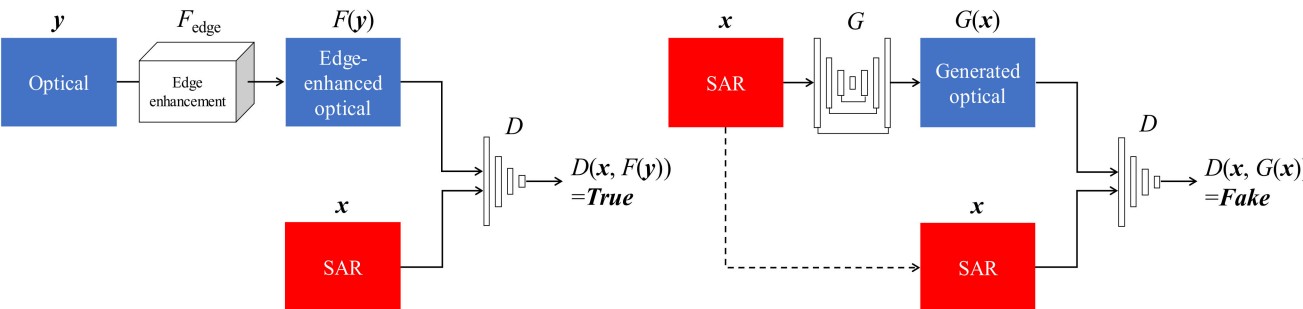

**Figure 3.** Structure of the training generator and discriminator model with an edge enhancement filter of the proposed method.

Since the objective of image translation is to find the corresponding points, important pieces of information in the generated image are the local image features (edges and corners). However, as shown in Figure 2, there were cases where the details were blurred and local features were lost in the cGAN. We solved this problem by proposing a method that applied an edge-enhancement filter, which adjusted pixel values of pixels along edges to emphasize edges, to the training data in advance to enable the cGAN network to learn the edges and corners more actively. The discriminator training is shown in Figure 3.

### 4. Finding Corresponding Points Using the Keypoint Detector and Descriptor

The keypoint-matching process is shown in Figure 4. As mentioned previously, the cGAN training model was used to obtain $G$, which generated images that were similar to the original optical images. Afterwards, keypoint matching was performed between the optical and generated optical images.

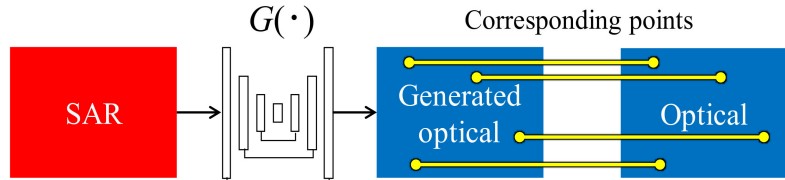

**Figure 4.** The keypoint-matching process. The trained $G$ obtained in Section 2 predicted optical images (generated optical images) from the input of SAR images.

Finding corresponding points consists of three major steps: keypoint detection, keypoint description, and keypoint matching. A typical process using SIFT [24], which is a major algorithm in keypoint matching, is outlined below. In the feature point detection step, the DoG (difference of Gaussian) image is used to detect tentative keypoints and the range (scale) is used for feature description. In the subsequent localization step, the sub-pixel positions are estimated by deleting the sub-pixels from the detected tentative keypoints that are not suitable for keypoints. In the feature description step, the direction of each feature point is determined from the bright gradient to obtain rotation-invariant features,

and 128-dimensional feature vectors are described for each feature point according to the direction and scale. In the final keypoint-matching step, two points in two images with similar feature vectors are extracted as corresponding points.

During the process of finding the corresponding points, false correspondences were also obtained. These false correspondences might decrease the registration accuracy; therefore, a process was required to remove them. In the case of matching between map-projected images, the difference between the two images could be regarded as scale, rotational, and translational transformation. Hence, the false corresponding points were removed based on the scale value and the gradient direction information of the correspondences [36]. Figure 5 shows the results of the corresponding points after eliminating the false correspondences.

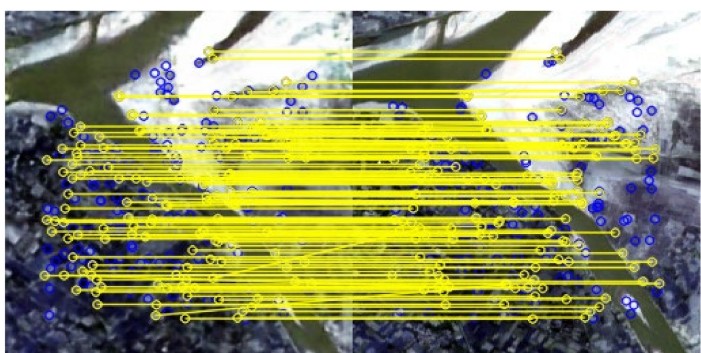

**Figure 5.** An example of a local correspondence between two images. The yellow lines are the connectors of corresponding points and the blue points are the points without corresponding points.

## 5. Evaluation of Keypoint Matching

### 5.1. Objective of the Evaluation

The objective of this experiment was to quantitatively evaluate the accuracy of the corresponding points to verify the effectiveness of the proposed method. Specifically, the accuracy of the positions where the corresponding points were extracted was evaluated. We used co-registered SAR and optical images for this experiment. In the evaluation scale, the average of the Euclidean distances $\bar{d}$ of the positions of the corresponding points in the SAR and optical image was used, which is given by the following:

$$\bar{d} = \frac{1}{n} \sum_{i}^{n} \| \boldsymbol{p}_i - \boldsymbol{p}_i' \|, \tag{4}$$

where $\boldsymbol{p}_i$ and $\boldsymbol{p}_i'$ are the positions of the corresponding points in the optical and generated optical image, respectively. If the registration was perfect, the corresponding points ideally had the same positions in each coordinate; hence, the closer the evaluation scale $\bar{d}$ was to 0, the higher the accuracy.

The results of the proposed method were compared with those of six other methods, which were as follows: (1) DLSC [14]—a method for finding corresponding points using dense local self-similarity; (2) HOPC [15]—a method for finding corresponding points using a histogram of orientated phase congruency (HOPC), which is based on the structural properties of images; (3) CFOG [16]—a method for finding corresponding points using CFOG, which is an extension of the pixel-wise HoG (histogram of Gaussian) descriptor; (4) Pix2pix [11]—a method that uses SIFT [24] for keypoint detection and description, and images obtained by a Pix2pix prediction; (5) Pix2pix + Edge Enhancement (EE)—a method that uses SIFT and edge-enhanced images obtained by applying an edge-enhancement filter to the Pix2pix prediction; and (6) Proposed—our method that uses SIFT for keypoint detection and description, and a discriminator trained by edge-enhanced images.

### 5.2. Environment of the Evaluation

5.2.1. The GAN Structure and Loss Function of the Experiment

Figure 6 shows the network structure used for learning. U-Net [37] was used for the generator, and PatchGAN [11,38] was used for the discriminator. U-Net and PatchGAN, which were used in the original Pix2pix, showed good and stable image translation results in its paper. Therefore, our proposed method was based on the Pix2pix network structure, and we mainly evaluated the effectiveness of the edge enhancement filter.

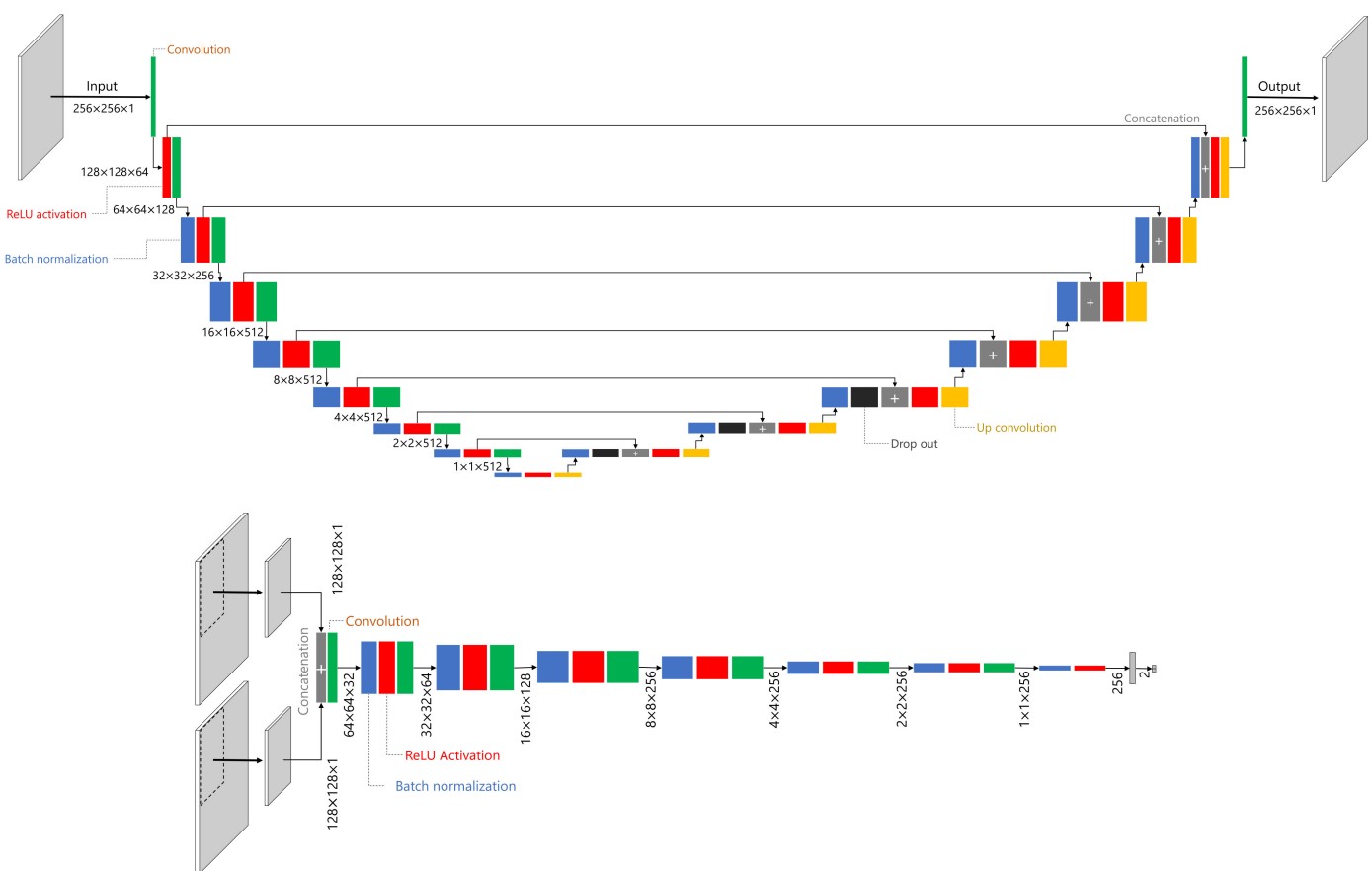

**Figure 6.** Structure of the generator and discriminator networks.

U-Net has skip structures in the layer, which makes it possible to pass information to the previous layer without loss before the convolution layer compresses the information. The number of down/up convolution layers was set to 8 ($= \log_2 256$), depending on the size of the training data, which was $256 \times 256$ pixels. To introduce randomness to the network, dropout (probability = 0.5) layers were added to three layers, as shown in Figure 6. The U-Net is considered appropriate for optical and SAR image translation because the proposed method preserves the edges and corners of the features, which are common in both optical and SAR images.

For the discriminator, the PatchGAN discriminator was used to discriminate the separated input images. The internal structure uses a general 7-layer convolutional encoder.

For the loss function, we adopted a function that uses $L_1$ norm, which is the same as that used in Pix2pix. The Pix2pix loss function is given by the following:

$$
\begin{aligned}
\mathcal{L}_{\text{Pix2pix}}(G, D) \quad &= \mathcal{L}_{\text{cGAN}}(G, D) + \lambda \mathcal{L}_{L1}(G) \\
&= \mathcal{L}_{\text{cGAN}}(G,\ D) + \lambda \mathbb{E}_{\boldsymbol{x},\ \boldsymbol{y}, \boldsymbol{z}}[\|\ \boldsymbol{y} - G(\boldsymbol{x},\ \boldsymbol{z})\ \|_1].
\end{aligned}
\tag{5}
$$

Referring to Equations (3) and (5), our proposed loss function is given by the following:

$$\mathcal{L}_{\text{Proposed}}(G, D) = \mathbb{E}_{\boldsymbol{y}}\left[logD\left(F_{\text{edge}}(\boldsymbol{y})\right)\right] + \mathbb{E}_{\boldsymbol{x},\boldsymbol{z}}[log(1 - D(\boldsymbol{x},\ G(\boldsymbol{x},\boldsymbol{z})))] + \lambda\mathbb{E}_{\boldsymbol{x},\ \boldsymbol{y},\boldsymbol{z}}\left[\parallel F_{\text{edge}}(\boldsymbol{y}) - G(\boldsymbol{x},\ \boldsymbol{z}) \parallel_1\right], \quad (6)$$

where $\lambda$ indicates a constant value, which was set to 10 [11] in this experiment, and $F_{\text{edge}}()$ indicates edge enhancement. Similar to Equation (2), the objective generator $G^{**}$ is given by the following:

$$G^{**} = arg\ \min_{G}\ \max_{D}\ \mathcal{L}_{\text{Proposed}}(G, D). \quad (7)$$

5.2.2. Dataset

The SEN1-2 dataset [39] was prepared from Sentinel-1 (SAR satellite) and Sentinel-2 (optical satellite) of the European Space Agency [40]. The dataset contains co-registered Sentinel-1 and Sentinel-2 image patches. Each item of the Sentinel-1 data has one 8-bit and $256 \times 256$-pixel channel (C-band, VV polarization), while that of Sentinel-2 data has three 8-bit and $256 \times 256$-pixel channels (red, green, and blue band). Their spatial resolution is 10 m per pixel. The "Urban", "Farm", and "Hill" areas were selected from the "spring" data in the SEN1-2 dataset due to their importance during a disaster event, and 2936, 3320, and 3224 image pairs were extracted, respectively. From these pairs, 300 sets each were selected as the test and validation data, respectively, and the remaining sets were used as training data. Figure 7 shows examples of the training, validation, and test image data for the experiment.

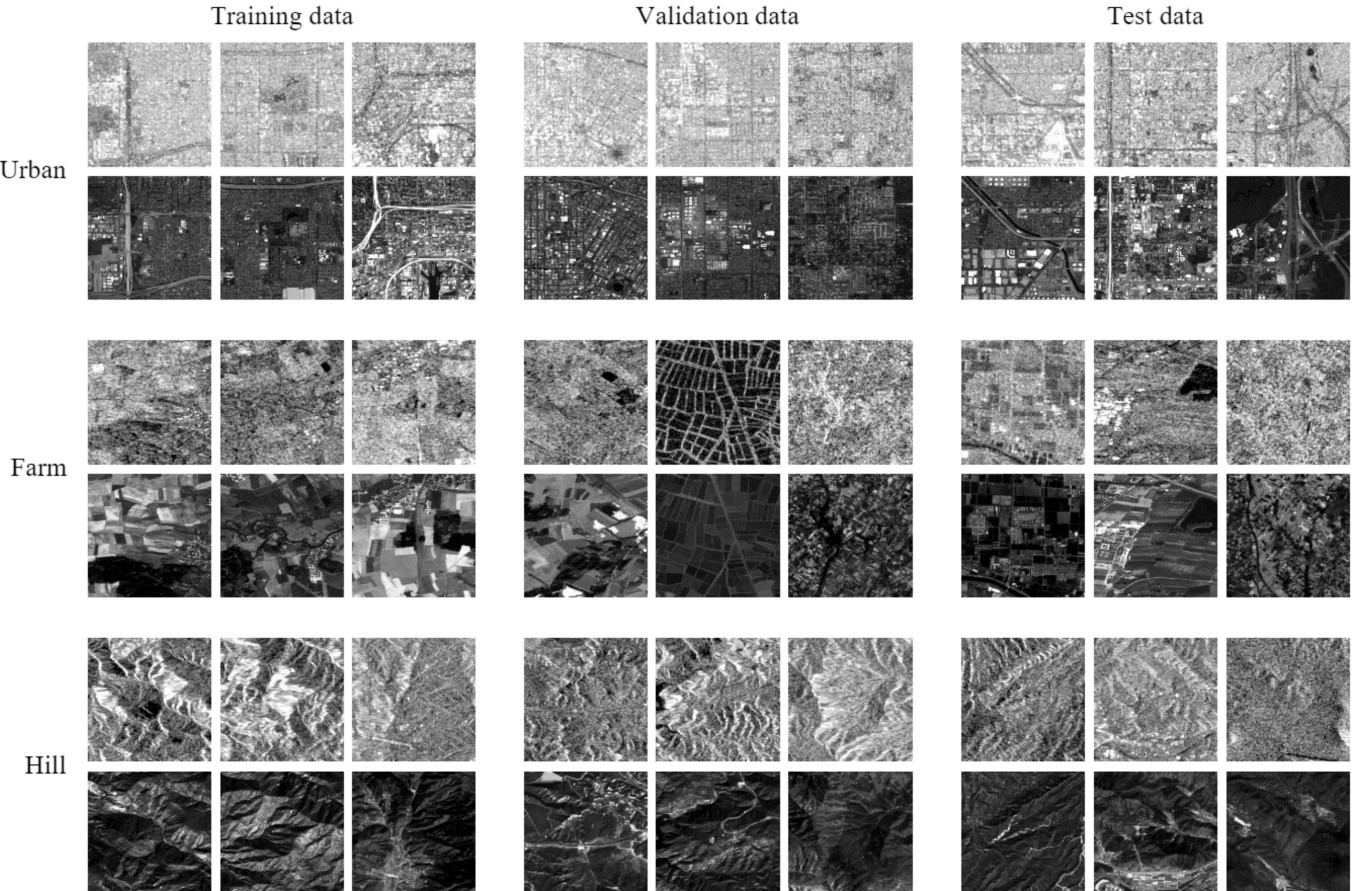

**Figure 7.** Examples of training, validation, and test images for the experiment. The upper half of each dataset represents SAR images, and the others represent optical images.

### 5.2.3. Implementation

The PatchGAN patch size was set to $128 \times 128$ pixels, with a batch size of 32. NVIDIA Tesla V100 GPU was used for parallel processing, and Adam [41] optimization was used to train the generator and discriminator, with a learning rate of $10^{-3}$.

Each training time was 36 h. The training time represented the time when the $L_1$ losses of the generators were low enough. The loss curves for each dataset are shown in Figure 8.

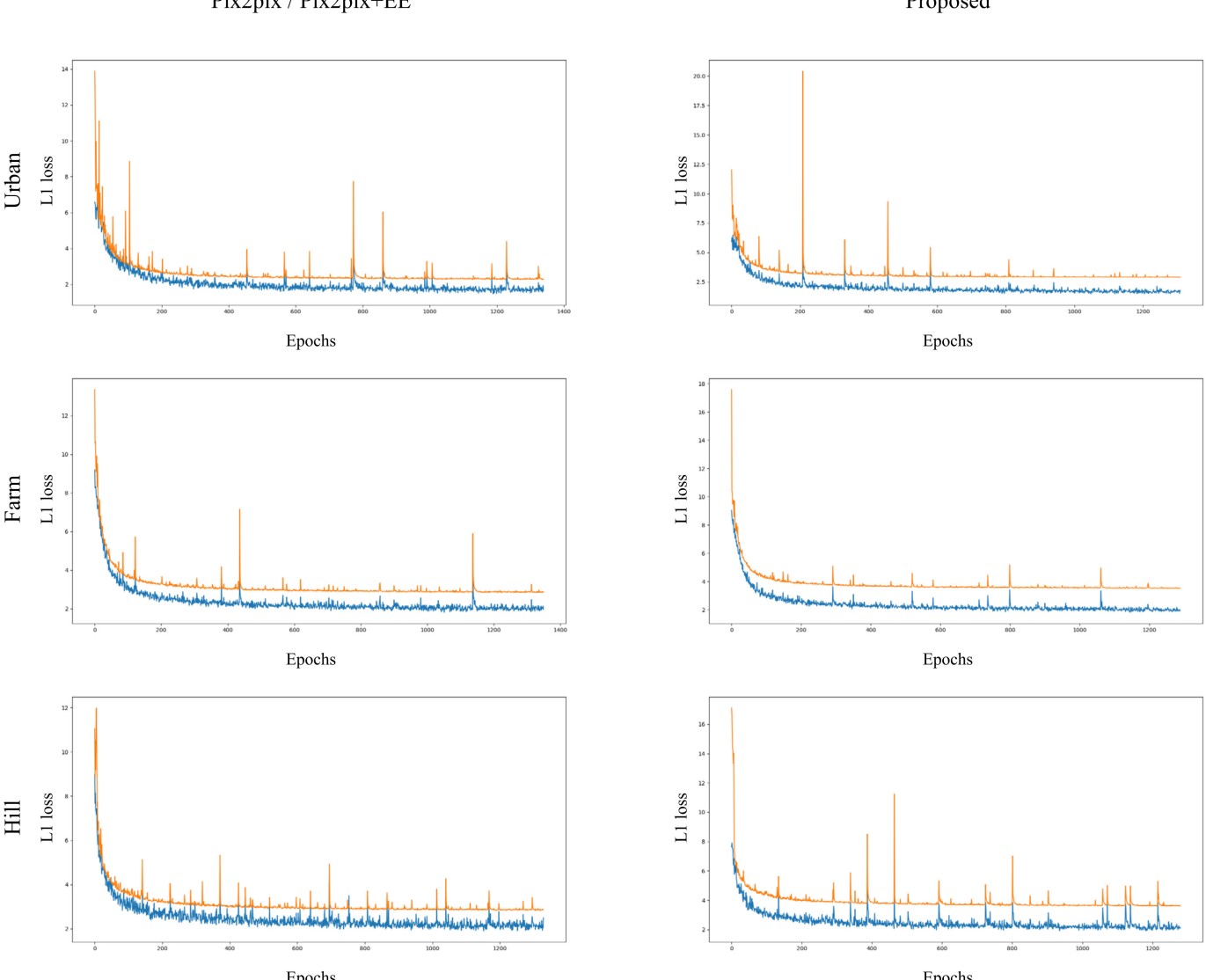

**Figure 8.** Loss curves of each data set. Blue lines show training, and orange lines show validation.

SIFT [24] was used as the keypoint detector and descriptor, and the parameters were the default values of OpenCV [42] version 3.4.3. To remove the false corresponding points [36], the threshold values of the scale and gradient direction were set to two octave layers and 5 degrees, respectively.

The template window sizes of the DLSC, HOPC, and CFOG were set to $100 \times 100$ pixels, and the distance threshold of the corresponding points was set to 1.5 pixels. This indicated that corresponding points with distances greater than 1.5 pixels we re considered outliers. The same threshold was applied to all the other methods.

An edge-enhancement filter based on the Laplacian filter was used, which is given by the following:

$$F_{edge}(\boldsymbol{I}) = \begin{bmatrix} -v & -v & -v \\ -v & 1+8v & -v \\ -v & -v & -v \end{bmatrix} \boldsymbol{I}, \tag{8}$$

where $v$ is a parameter for setting the strength of the edge enhancement. In this experiment, $v = 0.1$ was used because it revealed good results in preliminary experiments.

### 5.3. Result and Discussion

The generated optical images were properly translated when Pix2pix or the proposed method—which used GANs in their structure—were applied. The average, standard deviation and median values of the peak signal-to-noise ratio (PSNR) of 300 test images for Pix2pix were 22.70, 3.08, and 22.50 dB, respectively, whereas those of the proposed method were 23.01, 3.14, and 22.87 dB, respectively. PSNR is calculated as

$$PSNR(I_1, I_2) = 10 \times log_{10} \frac{MAX(I_1)^2}{MSE(I_1, I_2)}, \tag{9}$$

where $MAX(I_1)$ is the possible maximum value of image $I_1$, and $MSE(I_1, I_2)$ is the mean squared error between $I_1$ and $I_2$. $MAX(I_1)$ needs to equal $MAX(I_2)$.

Figure 9 shows the results of the SAR-to-optical image translation. It shows the input, output, and ground truth sets with the best PSNR values. Afterward, we evaluated the probability of these images for keypoint matching. Figure 10 shows an example of improvement by the proposed method. The proposed method improved the local blur and low reproducibility that were problems in the conventional cGANs.

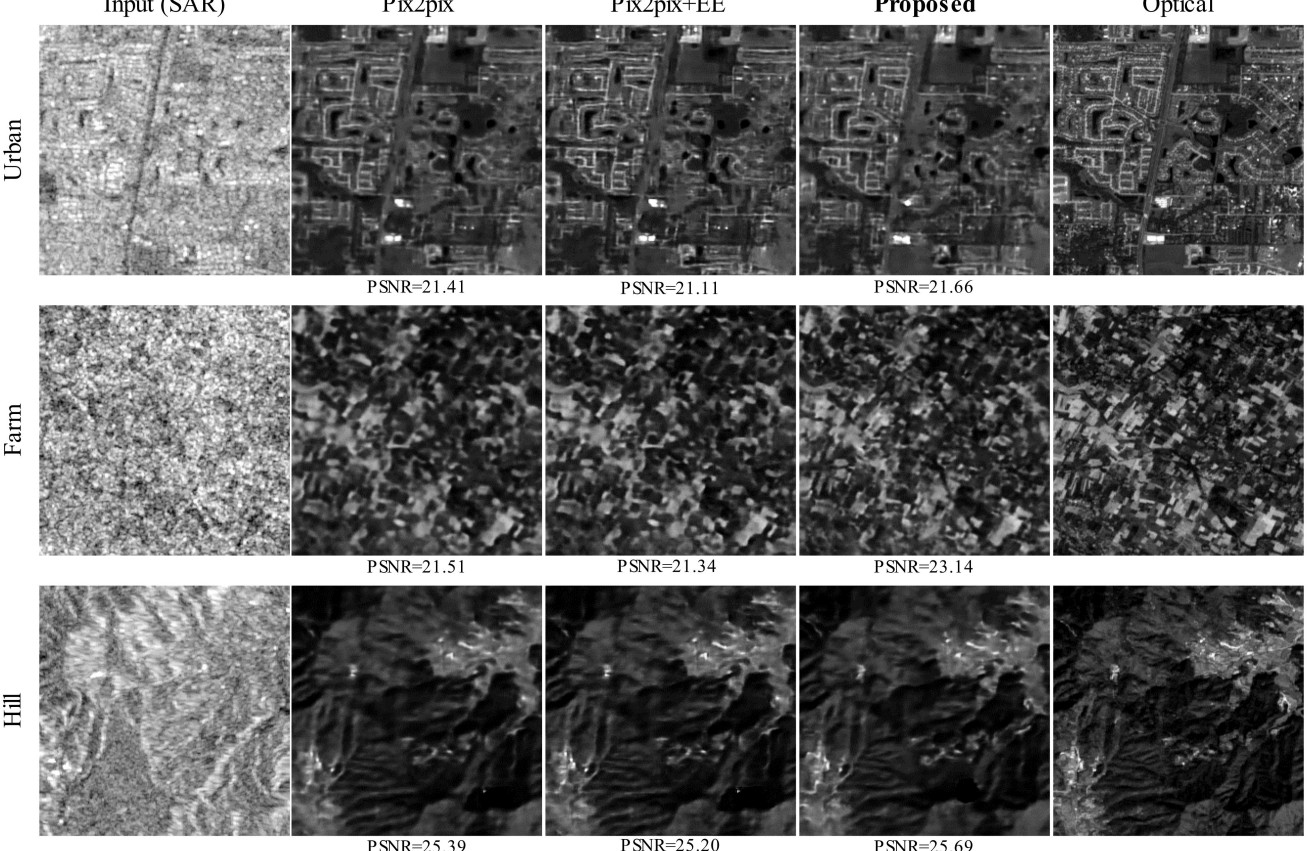

**Figure 9.** Results of SAR-to-optical image translation.

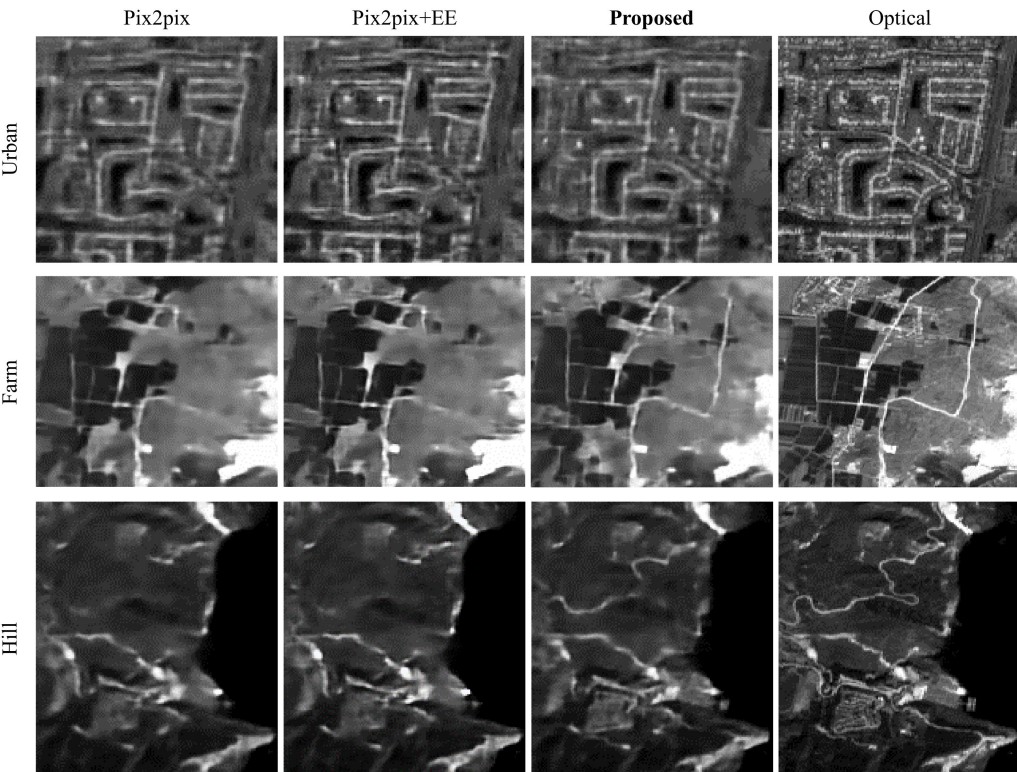

**Figure 10.** Improvement examples of image translation with our proposed method.

Tables 1 and 2 summarize the results of the quantitative evaluation experiments.

**Table 1.** Mean average (pixels) of the $\overline{d}$ in Equation (4).

|  | DLSC | HOPC | CFOG | Pix2pix | Pix2pix + EE | Proposed |
|---|---|---|---|---|---|---|
| Urban | 0.965 | 0.863 (10.6%) [3] | 0.838 (13.2%) | <u>0.564</u> [2] (<u>41.6%</u>) | 0.566 (41.3%) | **0.538** [1] **(44.2%)** |
| Farm | 0.847 | 0.847 (0%) | 0.700 (17.4%) | <u>0.675</u> (<u>20.3%</u>) | 0.676 (20.2%) | **0.569** **(32.8%)** |
| Hill | 1.128 | 1.052 (6.7%) | 1.114 (1.2%) | 0.668 (40.8%) | <u>0.667</u> (<u>40.9%</u>) | **0.577** **(48.9%)** |

[1] A bold number represents the best value in the row. [2] An underlined number represents the second best value in the row. [3] A value on each second row shows improving rate (%) compared to DLSC. This was calculated as $Improve\_rate_X = 100 \times (1 - Result_X / Result_{DLSC})$.

**Table 2.** Mean number of corresponding points (correct matches). The improvement score was larger in the "Farm" and "Hill" datasets.

|  | DLSC | HOPC | CFOG | Pix2pix | Pix2pix + EE | Proposed |
|---|---|---|---|---|---|---|
| Urban | 49.8 | **80.7** [1] **(62.0%)** | 63.0 (26.5%) | 67.6 (35.7%) | 63.5 (27.5%) | <u>78.5</u> [2] (<u>57.6%</u>) |
| Farm | 62.5 | <u>65.9</u> (<u>5.4%</u>) | 62.1 (−0.6%) | 57.4 (−8.2%) | 57.2 (−8.5%) | **101.1** **(61.8%)** |
| Hill | 17.2 | 45.2 (162.8%) | 22.6 (31.4%) | <u>57.5</u> (<u>234.3%</u>) | 56.4 (227.9%) | **85.0** **(394.2%)** |

[1] A bold number represents the best value in the row. [2] An underlined number represents the second best value in the row. [3] A value on each second row shows improving rate (%) compared to DLSC. This was calculated as $Improve\_rate_X = 100 \times (Result_X - Result_{DLSC}) / Result_{DLSC}$.

Table 1 shows the accuracy of the corresponding point detection. In the test dataset for each "Urban", "Farm", and "Hill" area, the proposed method had high precision with an average accuracy of 0.538 pixels, 0.569 pixels, and 0.577 pixels, respectively. The result

of the proposed method was more accurate than those of DLSC, HOPC, CFOG, Pix2pix, and Pix2pix + EE; the absolute accuracy was the highest in the "Urban" area. However, the improved accuracy of the "Farm" and "Hill" was higher than "Urban". This was because several objects had strong local features with sharp edges and corners, such as artificial structures, in urban areas. On the other hand, the local features of the "Farm" and "Hill" areas are weaker than those of the "Urban" area.

When edge enhancement was applied to the prediction result of Pix2pix in the Pix2pix + EE method, the accuracy was not as high as that of the proposed method, probably because the enhancement was equally applied to the artifacts generated in the prediction result. Hence, it was difficult to selectively enhance the effective local features.

Table 2 shows the number of detected corresponding points (correct matches). In the test dataset for each of the "Urban", "Farm", and "Hill" areas, the average scores of the proposed method were 78.5, 101.0, and 85.0 points, respectively. As the results in Table 1 indicate, the improvement score was larger in the "Farm" and "Hill" datasets. Although the number of corresponding points in the "Urban" dataset for HOPC was more than that of the proposed method, only the proposed method achieved more accuracy and several corresponding points.

Table 3 shows the inlier (correct matches) rates, in which our proposed method achieved a high inlier rate. For Pix2pix, Pix2pix + EE, and the proposed method, some outliers were already removed based on the scale value and the gradient direction of each keypoint [36]. This was the reason why the three methods achieved higher inlier rates than DLSC, HOPC, and CFOG.

**Table 3.** Mean inlier (correct matches) rates.

|  | DLSC | HOPC | CFOG | Pix2pix | Pix2pix + EE | Proposed |
|---|---|---|---|---|---|---|
| Urban | 0.437 | 0.703 (60.8%) [3] | 0.575 (31.6%) | 0.900 (105.9%) | 0.900 [2] (105.9%) | **0.913** [1] **(108.9%)** |
| Farm | 0.588 | 0.615 (4.6%) | 0.606 (3.1%) | 0.805 (36.9%) | 0.815 (38.6%) | **0.896 (52.4%)** |
| Hill | 0.155 | 0.396 (155.5%) | 0.200 (29.0%) | 0.809 (421.9%) | 0.808 (421.3%) | **0.880 (467.7%)** |

[1] A bold number represents the best value in the row. [2] An underlined number represents the second best value in the row. [3] A value on each second row shows improving rate (%) compared to DLSC. This was calculated as $Improve\_rate_X = 100 \times (Result_X - Result_{DLSC})/Result_{DLSC}$.

Figure 11 shows the results of the optical image generation and the subsequent feature-point matching. A sufficient number of correspondences were estimated, and it was confirmed that Pix2pix was effective for SAR-to-optical image translation. Although false correspondences remained, they could be removed using a robust estimation method, such as RANSAC [43].

As Figures 9 and 10 show, the Pix2pix + EE results indeed gave sharper images than those of Pix2pix and Proposed. However, the quantitative evaluation result showed better results than the others. The reason was that Pix2pix + EE enhanced not only effect image features but also artifacts. After all, the Pix2pix + EE results seemed sharper, but did not yield as good results.

Considering the "Urban" dataset in Figure 11 more closely, the center part of the generated optical image failed to translate in both methods. Even if fewer corresponding points were found in some parts of the images, it was possible to correctly calculate the total correspondences in a case where partial correspondence was correctly achieved in other areas since SIFT or other keypoint detection/description methods could calculate local features. Through this experiment, we confirmed that precise feature matching between the optical and generated optical images was possible.

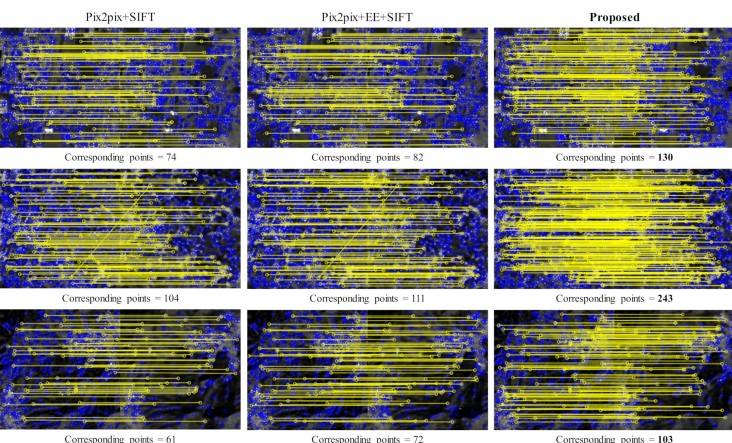

**Figure 11.** Results of keypoint matching. Yellow points connected by yellow lines mean corresponding points, and blue points mean other keypoints. No local feature correspondences were found in a collapsed area, but many corresponding points could be found in the remaining area.

A practical example of SAR-to-optical registration is shown in Figures 12 and 13. A SAR image was projected onto an optical image that had a rotation and translation formation. The projection was estimated by finding the corresponding points between the optical and generated optical images. We confirmed that high-precision image registration was achieved, even on the assumption of planarity, since this dataset covered an area that was not very wide. Alternatively, we could estimate a 3D projection of the corresponding keypoint if necessary [22].

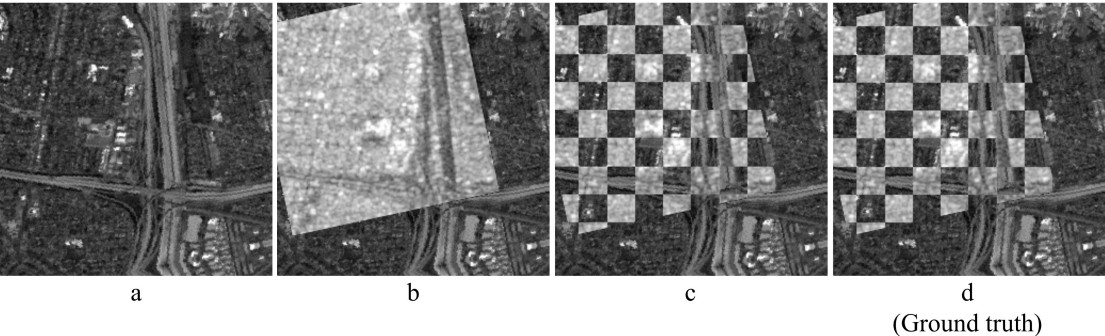

**Figure 12.** Example of SAR and optical image registration assuming planarity. To overlay a SAR image on the target optical image (**a**), after finding the local feature correspondences between the optical and generated optical images, the original SAR image was projected onto the optical image using an estimated homography transform (**b**). Half of the area was transparent, similar to a checkerboard pattern (**c**), and ground truth, which was registered by accurate DEM (**d**).

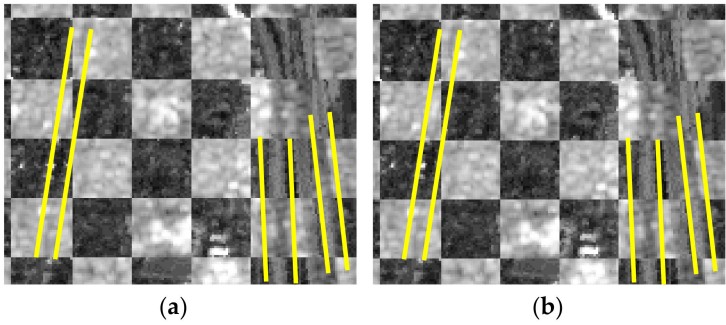

**Figure 13.** (**a**,**b**) are Close-up of a part of Figures c and d in Figure 12. Yellow lines were drawn along the roads to demonstrate the registration accuracy. It could be confirmed that image registration between SAR and optical images was performed with high accuracy.

A limitation of the proposed method was that it was hard to apply this method to very high spatial resolution images; we confirmed this with 3-m/pixel imagery by Cosmo-SkyMed [44]. That was because layovers in SAR images prevented training datasets from accurate image registration. We confirmed that the proposed method worked well in middle resolution images (about 10 m/pixel). Considering the balance between the size of the observation area and its detail, the dataset of 10 m/pixel used in this evaluation was considered to be an appropriate spatial resolution dataset.

## 6. Conclusions

In this paper, we proposed a method for translating SAR images to optical images using a GAN so that a keypoint-matching algorithm could be applied to multimodal images. By applying this method for SAR-to-optical image translation, we performed keypoint-matching on the monomodal images. Through quantitative evaluations of the keypoint-matching accuracy, we confirmed that the proposed method could achieve accurate keypoint matching between the optical and generated optical images using a cGAN. In the translation from optical to generated optical images using the cGAN, the local features could not be obtained, and the corresponding points could not be established due to blurring. Therefore, an improvement was achieved in our proposed method by applying the edge-enhancement filter to the training data for the discriminator and training the generator to actively learn the local features. Furthermore, we conducted a comparative experiment and confirmed that our proposed method was effective for finding local corresponding points.

**Author Contributions:** Conceptualization, H.T., A.D. and I.K.; methodology, H.T., M.S. and I.K.; software, H.T.; validation, H.T., N.O. and F.I.; formal analysis, H.T.; investigation, H.T.; resources, A.D. and I.K.; data curation, H.T. and H.I.; writing—original draft preparation, H.T., H.I. and N.O.; writing—review and editing, H.T., A.D. and I.K.; visualization, H.T. and N.O.; supervision, A.D., Y.K. and I.K.; project administration, I.K.; funding acquisition, I.K. All authors have read and agreed to the published version of the manuscript.

**Funding:** This work was supported by JST CREST under Grant JPMJCR16E3; and Grant-in-Aid 456 for JSPS Fellows under Grant JP19J11514.

**Institutional Review Board Statement:** Not applicable.

**Informed Consent Statement:** Not applicable.

**Data Availability Statement:** Not applicable.

**Acknowledgments:** Drone photography was conducted in cooperation with DRONEBIRD by Crisis Mappers Japan (NPO), a disaster drone rescue team. (http://dronebird.org/, accessed on 14 March 2022.)

**Conflicts of Interest:** The authors declare no conflict of interest.

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
