# Peer review of "Use of a DNN-Based Image Translator with Edge Enhancement Technique to Estimate Correspondence between SAR and Optical Images"

_applsci, doi:10.3390/app12094159_

Round 1

Reviewer 1 Report

The paper discusses a minor improvement of the Pix2pix algorithm which matches synthetic aperture radar (SAR) and visual images. The improvement consists of applying edge enhancement to the visual image before training the underlying cognitive generative adversarial network. After a clear overview of the problem the algorithms are discussed, followed by the experimental results. Apart from some minor errors, the presentation is well written.

Minor remarks:

Lines 89-97: refer to sections by their numbers, do not include section titles.
Line 101: "based on the difference in their frequency spectra" -- Maybe similarity, correspondence ?
Line 110: "these methods consider ... " what does this sentence mean?
Line 241: "... as the original Pix2pix uses." ???
Line 251: "depending on the size"  Is this value was set to 8 because of the typical image sizes in the experiment? Please be more punctuate.
Line 254: "the proposed" => "it" ?
Lines 264--267: I do not see the relevance of this sentence.
Line 278: there are two "training" words here
Line 279-281: It would be better not to show the RGB images, rather the converted grayscale images only. It is not clear what these images are. I expect one SAR image and one single grayscale visual data.
Lines 322-323: These lines belong to Table 1
Line 324: The start of this sentence is missing.
Line 329: Rewrite.. Which ones are higher and which ones are high? 
Lines 342-343: Should go to Table 2
Line 344: Missing the start of the sentence.
Line 348: this is a footnote to Table 2
Line 349: the beginning of this sentence is missing
Line 373: "not only effective" => "not only effect" 
Line 374: "not so good results" => "did not yield as good results"
Line 503: Janua*ry*

Author Response

Thank you very much for your careful review. We appreciate your competent and helpful suggestions for this paper.

Our responses for your comments are listed as follows: 

  • Lines 89-97: refer to sections by their numbers, do not include section titles.

[Response] Thank you for pointing this out. We have made the correction according to your comment.

  • Line 101: "based on the difference in their frequency spectra" -- Maybe similarity, correspondence ?

[Response] Thank you for pointing this out. It was indeed a difficult expression to understand, so we changed it to "the similarities of their edges and corners."

  • Line 110: "these methods consider ... " what does this sentence mean?

[Response] It was indeed a difficult expression to understand, so we have corrected it.

  • Line 241: "... as the original Pix2pix uses." ???

[Response] Thank you for pointing this out. The proposed method was developed based on pix2pix [11], so it uses some of the same parameters and structures as pix2pix. We have removed the description because it is indeed misleading and unnecessary.

  • Line 251: "depending on the size"  Is this value was set to 8 because of the typical image sizes in the experiment? Please be more punctuate.

[Response] Thank you very much for your suggestion. The reason we set the "number of down/up convolution layers" to 8 this time is because the size of the input image data was 256x256. Since the convolution layer halves the data size each time, 8 is the maximum value for "number of down/up convolution layers" since 2^8=256.

  • Line 254: "the proposed" => "it" ?

[Response] Thank you for your comment. It should have been “the proposed method.” We've modified.

  • Lines 264--267: I do not see the relevance of this sentence.

[Response] Thank you for pointing this out. It is true that the same thing has been explained in other texts, and we do not feel that it should be mentioned again in this chapter, so we deleted the paragraph.

  • Line 278: there are two "training" words here

[Response] Thank you for pointing this out. The "training" was a mistake and has been corrected into “test.”

  • Line 279-281: It would be better not to show the RGB images, rather the converted grayscale images only. It is not clear what these images are. I expect one SAR image and one single grayscale visual data.

[Response] Thank you for your comment. We have replaced them with the grayscale satellite images. The reason we include three images for each data set is to show that data variability is guaranteed, and we believe that three images should be included in each figure because we think it is better than showing only one image.

  • Lines 322-323: These lines belong to Table 1

[Response] Thank you very much for your comments. Lines 322-323 "Table 1, Table 2, and Table 3 summarize the results of the quantitative evaluation experiments."  are sentences describing Tables 1, 2 and 3. However, the sentence immediately after that is a sentence explaining only Table 1, so it was difficult to understand. Therefore, we have separated the paragraphs and clarified them.

  • Line 324: The start of this sentence is missing.
  • Line 344: Missing the start of the sentence.
  • Line 349: the beginning of this sentence is missing

[Response] Thank you for pointing them out, but when we checked that line of text in our environment, we did not see any incomplete sentences. Perhaps the cross-referencing function in WORD was not working properly and "Table X" was not displayed. If we have misunderstood your comment, we would be happy to point it out again.

  • Line 329: Rewrite.. Which ones are higher and which ones are high? 

[Response] Thank you for pointing this out. In response to your comment, we have made a correction to clarify the relationship between large and small.

  • Lines 342-343: Should go to Table 2

[Response] Thank you for pointing this out. For clarity, we have added this sentence to the caption of Table 2.

  • Line 348: this is a footnote to Table 2

[Response] Thank you for pointing this out. The style of the text was incorrect and has been corrected.

  • Line 373: "not only effective" => "not only effect" 
  • Line 374: "not so good results" => "did not yield as good results"
  • Line 503: January

[Response] Thank you for pointing them out. We have followed your comments and made the corrections.

Reviewer 2 Report

A method for translating SAR images to optical images using GAN is proposed. Keypoint matching on the images is performed by applying this method for SAR-to-optical image translation. The manuscript is well-organized and the effectiveness of the method are well supported by the comparative analysis. Several questions:

  1. In the keypoint matching process, how many keypoints are typically selected to be matched? What’s the basis for the selection?
  2. In line 180“……they can be obtained by applying an edge-enhancement filter.” Some comments are needed to explain the edge-enhancement filter.
  3. In line 402-405, it is mentioned that “We confirmed that the proposed method works well in middle resolution images (about 10 m/pixel). Considering the balance between the size of the observation area and its detail, the dataset of 10 m/pixel used in this evaluation is considered to be an appropriate spatial resolution dataset.” If the resolution is lower or higher than 10 m/pixel, how would the method proposed in this manuscript perform? The balance should be elaborated.

Author Response

Thank you very much for your careful review. We appreciate your competent and helpful suggestions for this paper.

Our responses for your comments are listed as follows: 

  • In the keypoint matching process, how many keypoints are typically selected to be matched? What’s the basis for the selection?

[Response] Thank you for your question. The keypoint detection algorithm used in this work, such as SIFT [23], is not an algorithm that detects a specified number of keypoints, but one that quantitatively defines the "feature value" then selects the keypoint after thresholding. The threshold value is often set at around 0.70.

  • In line 180 “……they can be obtained by applying an edge-enhancement filter.” Some comments are needed to explain the edge-enhancement filter.

[Response] Thank you for your comment. We added some explanations about edge-enhancement filters. There are some edge-enhancing methods based on some edge-detection methods. We used Laplacian based edge enhancement in the experiment, and the detail is shown in 5.2.3.

  • In line 402-405, it is mentioned that “We confirmed that the proposed method works well in middle resolution images (about 10 m/pixel). Considering the balance between the size of the observation area and its detail, the dataset of 10 m/pixel used in this evaluation is considered to be an appropriate spatial resolution dataset.” If the resolution is lower or higher than 10 m/pixel, how would the method proposed in this manuscript perform? The balance should be elaborated.

[Response] Thank you for your question. If the spatial resolution is higher than 10 m, this method works because the effect of layover is minimal. However, the finer the spatial resolution, the more effective it is for understanding the situation on the ground. We believe that a spatial resolution of about 10 m/pixel is appropriate if you are choosing from among satellite images that are generally available today.

Reviewer 3 Report

Please kindly add the improve accuracy percentage as it is much easier to understand.

Line 33-35, how much deterioration would that be in percentage relative to height of the satellites' flights?

Line 168-168, how did you do that co-registered image pairs, by which method?

In table 2 and 3, you should also show how much percentage your proposed method is better than others?  You should have anther column to provide that improve percentage.

Line 397-398, high accuracy, how high is it?  Can you provide the exact accuracy number?

Line 400-405, you meant you already tested that and it gave a poor result.  What satellite's imageries you used to test that?  Please kindly elaborate more!

Author Response

Thank you very much for your careful review. We appreciate your competent and helpful suggestions for this paper.

Our responses for your comments are listed as follows: 

  • How many percentage of the matching accuracy? (Line. 23 in the PDF.)
  • Please kindly add the improve accuracy percentage as it is much easier to understand.
  • In table 2 and 3, you should also show how much percentage your proposed method is better than others?  You should have anther column to provide that improve percentage.

[Response] Thank you for your suggestions, and we added the improved accuracy percentage. It certainly makes it easier to understand.

  • Line 33-35, how much deterioration would that be in percentage relative to height of the satellites' flights?

[Response] Thank you for your question. The term "deterioration" here refers to scattering and attenuation by clouds and other water vapor. Unless it is raining very hard, the radar of SAR will not scatter or attenuate. Therefore, there is no direct relationship with altitude.

  • Very good point! (Line. 50-53)

[Response] Thank you very much.

  • Line 168-168, how did you do that co-registered image pairs, by which method?

[Response] Thank you for your question. We used the SEN1-2 dataset [39], which is a co-registered SAR and optical image pair dataset. The authors [39] achieved co-registering process with DEMs.

  • Line 397-398, high accuracy, how high is it?  Can you provide the exact accuracy number?

[Response] Thank you for your comments. Quantitative accuracy has been verified in tables 1, 2 and 3, so this Figure 13 shows qualitative accuracy. The purpose of this figure is to show that it is visually accurate, since it can be confirmed that the roads in the SAR and optical images are correctly connected.

  • Line 400-405, you meant you already tested that and it gave a poor result.  What satellite's imageries you used to test that?  Please kindly elaborate more!

[Response] Thank you for your question. We used Cosmo-SkyMed imagery (about 3 m/pixel), and we've added the information.

Reviewer 4 Report

The paper proposes a method for performing SAR-to-optical-image translation using GAN, so the algorithm can be applied to multimodal images. The results suggested that the proposed method can accurately estimate the corresponding points between SAR and optical images.

But, the authors contributions are not sustain by some scientific support and these aren't very clearly defined. The paper looks like a story. It is hard to follow. The organization and the flow of the paper need improvements.

In introduction should be a state-of-the-art and and clearly highlighted the author's contributions. Some results should be briefly mentioned.

Figures 1 and 2 are the authors results? If not, it should be a reference for each one. These figures should be in the results section. In introduction, there are not showed the results, even these are preliminary.

For the rest of the paper, it should be a system design, a methods description with scientific support. This is mandatory.

The methods used for comparison should be described a little bit. 

Author Response

Thank you very much for your careful review. We appreciate your competent and helpful suggestions for this paper.

Our responses for your comments are listed as follows: 

  • The paper proposes a method for performing SAR-to-optical-image translation using GAN, so the algorithm can be applied to multimodal images. The results suggested that the proposed method can accurately estimate the corresponding points between SAR and optical images. But, the authors contributions are not sustain by some scientific support and these aren't very clearly defined. The paper looks like a story. It is hard to follow. The organization and the flow of the paper need improvements.
    In introduction should be a state-of-the-art and clearly highlighted the author's contributions. Some results should be briefly mentioned.

[Response] Thank you for your comments. Indeed, we thought that the contributions of the authors were not clearly stated, so we have made some corrections in the introduction. (See the last three paragraphs in Introduction.) The authors' related previous work [10] and the contribution of this paper were not clearly separated, which made it difficult to understand.

  • Figures 1 and 2 are the authors results? If not, it should be a reference for each one. These figures should be in the results section. In introduction, there are not showed the results, even these are preliminary.

[Response] Thank you for your comment. First of all, this Figure 1 is original by the authors, as it was created to give readers an overview of the proposed method referring to our previous study [10]. Figure 2 is an image we prepared to show the main problem of this paper, CGAN blur, and is an image we obtained in our preliminary experiments. However, as you said, the reference was necessary because the image was obtained using pix2pix, which is one of the CGANs. Thank you very much.

  • For the rest of the paper, it should be a system design, a methods description with scientific support. This is mandatory.

[Response] Thank you for pointing this out. It is true that, as you pointed out, we did not make the contributions of this paper clear in the introduction, so we have revised the introduction (see the second paragraph from the end of the introduction.) After that, however, we explained the details of the system, citing important papers as appropriate, and conducted quantitative and qualitative comparison experiments with other methods based on published data sets and indices defined by mathematical formulas to prove the effectiveness of the proposed method. We believe this paper has sufficient scientific support.

  • The methods used for comparison should be described a little bit. 

[Response] Thank you for pointing this out. The comparison methods are briefly explained in the second paragraph of section 5.1, along with references. We have added an explanation of HOPC since it was missing.

Round 2

Reviewer 3 Report

Thank you very much for your replies, excellent works!!!

Reviewer 4 Report

The revised paper can be published in the present form.